# Gill developmental program in the teleost mandibular arch

**Mathi Thiruppathy[1], Peter Fabian[1], J Andrew Gillis[2,3], J Gage Crump[1]***

[1]Eli and Edythe Broad California Institute for Regenerative Medicine Center for Regenerative Medicine and Stem Cell Research, Department of Stem Cell Biology and Regenerative Medicine, University of Southern California Keck School of Medicine, Los Angeles, United States; [2]Marine Biological Laboratory, Woods Hole, United States; [3]Department of Zoology, University of Cambridge, Cambridge, United Kingdom

**Abstract** Whereas no known living vertebrate possesses gills derived from the jaw-forming mandibular arch, it has been proposed that the jaw arose through modifications of an ancestral mandibular gill. Here, we show that the zebrafish pseudobranch, which regulates blood pressure in the eye, develops from mandibular arch mesenchyme and first pouch epithelia and shares gene expression, enhancer utilization, and developmental *gata3* dependence with the gills. Combined with work in chondrichthyans, our findings in a teleost fish point to the presence of a mandibular pseudobranch with serial homology to gills in the last common ancestor of jawed vertebrates, consistent with a gill origin of vertebrate jaws.

## Editor's evaluation

This is an interesting and important paper that investigates pseudobranch development in zebrafish in the context of seeking evidence for a proposed gill arch origin for the vertebrate jaw. It provides data that supports that the pseudobranch is derived from the mandibular arch and that the pseudobranch is a segmental homolog of the gills providing strong support for the classic gills-to-jaws hypothesis.

*For correspondence:
gcrump@usc.edu

**Competing interest:** The authors declare that no competing interests exist.

## Introduction

Gills are the major sites of respiration in fishes. They are composed of a highly branched system of primary and secondary filaments, housing blood vessels, a distinct type of cellular filament cartilage, pillar cells (specialized endothelial cells), and epithelial cells maintaining ionic balance. In teleost gills, two rows of filaments are anchored to a prominent gill bar skeleton. Both the filaments and supportive gill bars develop from the embryonic pharyngeal arches that consist of mesenchyme of neural crest and mesoderm origin and epithelia of endodermal and ectodermal origin (***Fabian et al., 2022***; ***Mongera et al., 2013***). The third and more posterior arches generate gills in most fishes. The second (hyoid) arch also forms a hemibranch (one row of gill filaments) in the jawless lamprey fish (***Dohrn, 1882***; ***Gaskell, 1908***), in cartilaginous and various non-teleost fishes (e.g. coelacanth, lungfishes, sturgeon, and gar), but not in teleost fishes (***Goodrich, 1930***; ***Jollie, 1962***). A classical theory for the origin of jaws posits that an ancestral gill support skeleton in the mandibular arch was repurposed for jaw function (***Gegenbaur et al., 1878***). However, extant agnathans (the cyclostomes lamprey and hagfish) lack a mandibular gill (***Cole, 1905***; ***Mallatt, 1996***), and fossil evidence for ancestral vertebrates with a mandibular gill is scant. Whereas exceptional soft tissue preservation of *Metaspriggina walcotti* from

the Cambrian Burgess Shale had suggested a dorsoventrally segmented cartilaginous gill bar in the presumptive mandibular arch, gill filaments were not observed (*Morris and Caron, 2014*).

The pseudobranch is an epithelial structure located just behind the eye that has been proposed to regulate ocular blood pressure and/or have an endocrine function (*Jollie, 1962*). While it shares an anatomical resemblance to gill filaments and is found in many jawed fishes (*Dohrn, 1882*), its embryonic arch origins remain debated (*Miyashita, 2016*). In parallel work in little skate, we identify a mandibular arch origin of the pseudobranch in chondrichthyans (*Hirschberger and Gillis, 2022*), which form one branch of jawed vertebrates. Whether the mandibular origin of the pseudobranch is conserved across vertebrates, including bony fishes, remained unknown. Another important question is whether the pseudobranch and gills can be considered serially homologous, i.e., representing morphologically related structures that arise through shared developmental and genetic mechanisms. Through lineage tracing and genetic analyses in zebrafish here, and lineage analysis in skate (*Hirschberger and Gillis, 2022*), we infer that the pseudobranch is a mandibular arch-derived serial homolog of the gills that was present in at least the last common ancestor of jawed vertebrates.

## Results

In zebrafish, the pseudobranch is located anterior to the gill filaments and connected to the eye via the ophthalmic artery (*Figure 1a, c*), as described for other fishes (*Laurent and Dunel-Erb, 1984*). The pseudobranch appears in histological sections as a small bud behind the eye at 4 days post-fertilization (dpf) (*Figure 1b*). Examination of *Sox10:Cre; acta2:loxP-BFP-Stop-loxP-dsRed* zebrafish shows this bud to be composed of a core of Cre-converted dsRed+ neural crest-derived cells ensheathed by unconverted BFP+ epithelia (*Figure 1—figure supplement 1a*). The position of this bud corresponds to *kdrl:mCherry* labeling of a branch of the first aortic arch that likely gives rise to the ophthalmic artery (*Figure 1—figure supplement 1b*). At 17 dpf, the pseudobranch is composed of five distinct filaments that resemble the primary gill filaments, with the five filaments merging to form a single pseudobranch by adult stages (90 dpf) (*Figure 1b*). Alcian Blue staining reveals that the adult pseudobranch contains five cartilage rods, reflecting the five fused filaments, with this cartilage resembling the specialized filament cartilage seen in the gills (*Figure 1d*; *Fabian et al., 2022*).

To determine from which arch the pseudobranch arises, we performed short-term lineage tracing using a photoconvertible *sox10:kikGR* reporter expressed in neural crest-derived mesenchyme. Photoconversion of dorsal first arch mesenchyme at 1.5 dpf labeled the pseudobranch mesenchymal bud at 3.5 dpf, as well as the palatoquadrate cartilage, a known first arch derivative; photoconversion of dorsal second arch mesenchyme did not label the pseudobranch (*Figure 1e*). To trace the epithelial origins of the pseudobranch, we performed short-term lineage tracing using *fgf10b:nEOS*, in which the photoconvertible nuclear-EOS protein is expressed in endodermal pouch epithelia (*Figure 1—figure supplement 1c*). Photoconversion of first pouch endoderm at 1.5 dpf labeled pseudobranch epithelia at 5 dpf (*Figure 1f*; *Figure 1—figure supplement 1e*), similar to labeling of first gill filament epithelia after photoconversion of third pouch endoderm (*Figure 1—figure supplement 1f*). We also confirmed endodermal origin of *cdh1:mlanYFP+* pseudobranch epithelia by 4OH-tamoxifen-mediated conversion of early endoderm in *sox17:CreERT2; ubb:loxP-Stop-loxP-mCherry* zebrafish (*Figure 1—figure supplement 1d*). The pseudobranch therefore arises from mandibular arch neural crest-derived mesenchyme and first pouch endodermal epithelia.

In skate, the pseudobranch and gills share expression of *foxl2*, *shh*, *gata3*, and *gcm2* (*Hirschberger and Gillis, 2022*). To test whether this reflects shared gene regulatory mechanisms indicative of serial homology, we examined activity of several gill-specific enhancers (*Fabian et al., 2022*). At 5 dpf, the gata3-p1 enhancer drives GFP expression in the growing tips of both the pseudobranch and gill buds (*Figure 2a*). At 14 dpf, the ucmaa-p1 enhancer, active in gill filament but not hyaline cartilage in the face, drives GFP expression in both pseudobranch and gill filament cartilage (*Figure 2b*), as seen for endogenous expression of *ucmaa* (*Figure 3—figure supplement 1a*). In our single-cell chromatin accessibility analysis of neural crest-derived cells (*Fabian et al., 2022*), we also identified an *irx5a* proximal enhancer selectively accessible in pillar cells, a specialized type of endothelial cell in the gill secondary filaments (*Figure 3—figure supplement 2a*). At 13, 20, and 60 dpf and one-year-old adult fish, the irx5a-p1 enhancer drives GFP expression in pillar cells of the pseudobranch and gills (*Figure 2c*; *Figure 3—figure supplement 2b, c*). These findings show that cells with similar gene expression and cis-regulatory architecture are present in both the pseudobranch and gills of zebrafish.

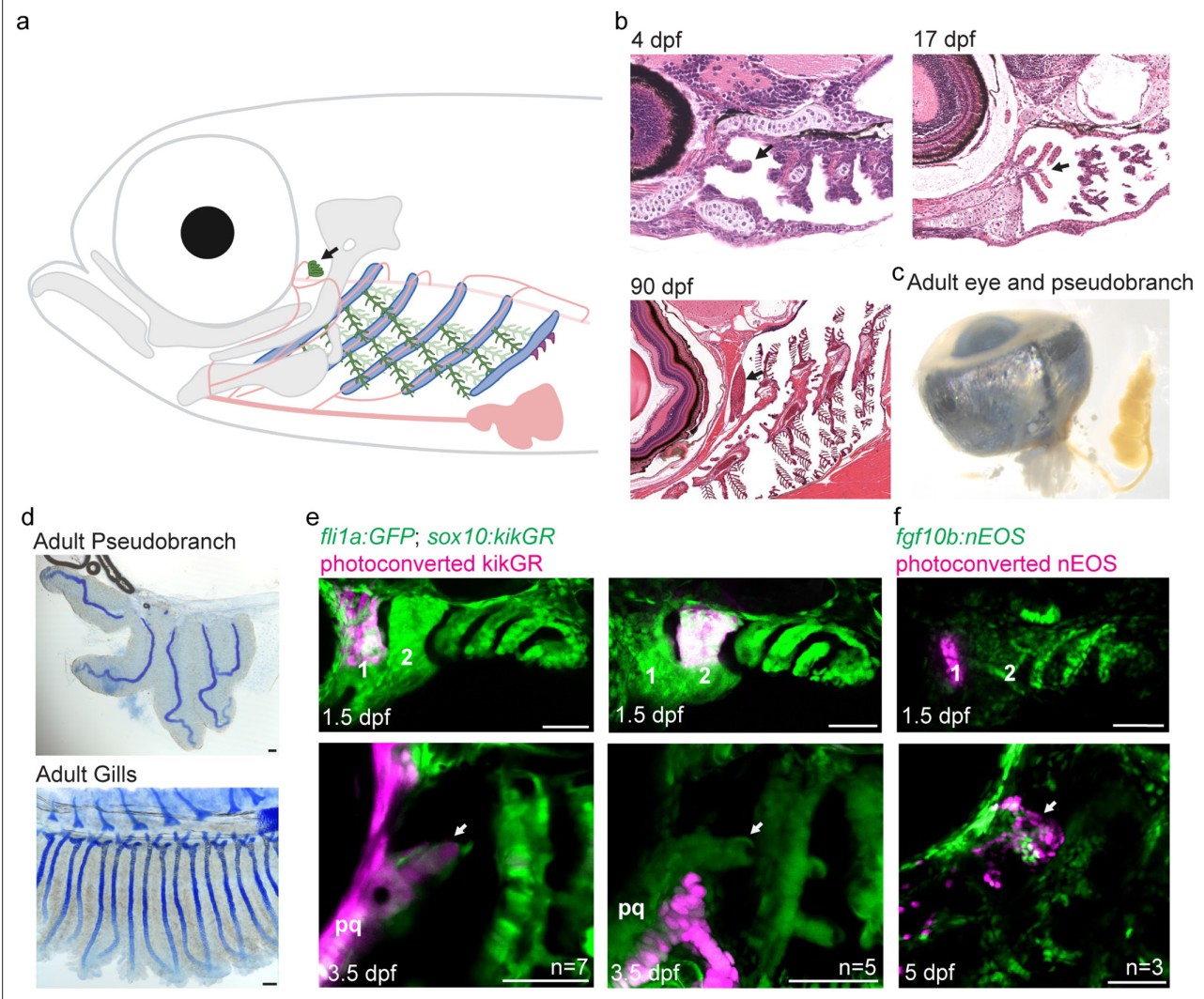

**Figure 1.** The zebrafish pseudobranch derives from mandibular arch mesenchyme and first pouch epithelia. (**a**), Schematic showing the pseudobranch (arrows), gill filaments (branched green structures) connected to gill bars (blue), teeth (purple), vasculature (pink), and jaw and jaw-support skeleton (gray). (**b**) Hematoxylin and Eosin-stained sections show emergence of the pseudobranch bud at 4 dpf (adapted from https://bio-atlas.psu.edu/zf/view.php?atlas=5&s=41), five filaments at 17 dpf (adapted from https://bio-atlas.psu.edu/zf/view.php?atlas=65&s=1738), and the fused pseudobranch at 90 dpf (adapted from https://bio-atlas.psu.edu/zf/view.php?atlas=29&s=312). (**c**) Dissected adult pseudobranch shows the ophthalmic artery connecting it to the eye. (**d**) Alcian staining shows five cartilage rods in the pseudobranch and similar cartilage in gill primary filaments. (**e**) Photoconverted kikGR-expressing mesenchyme (red) from the dorsal first arch (numbered) at 1.5 dpf contributes to the palatoquadrate cartilage (pq) and pseudobranch mesenchyme (arrow) at 3.5 dpf. Photoconverted dorsal second arch cells do not contribute to the pseudobranch. In green, *fli1a:GFP* labels the vasculature and neural crest-derived mesenchyme, with mesenchyme also labeled by unconverted *sox10:kikGR*. (**f**) In *fgf10:nEOS* embryos, photoconversion of first pouch endoderm (numbered) at 1.5 dpf labels the pseudobranch epithelium (arrow) at 5 dpf. n numbers denote experimental replicates in which similar contributions were observed. Scale bars, 50 μm.

The online version of this article includes the following figure supplement(s) for figure 1:

**Figure supplement 1.** Development of zebrafish pseudobranch and lineage analysis of gill filament epithelia.

Zebrafish mutant for *gata3* fail to form gill buds (*Sheehan-Rooney et al., 2013*), and single-cell chromatin accessibility analysis of neural crest-derived cells had implicated *gata3* and *gata2a* in development of gill filament cell type differentiation (*Fabian et al., 2022*). We find that *gata3* and *gata2a* are prominently expressed in both the developing pseudobranch and gill buds at 3 and 5 dpf (*Figure 3a*; *Figure 3—figure supplement 1b, c*). The pseudobranch is also much reduced in *gata3* mutants at 5 dpf, with fewer neural crest-derived cells labeled by *Sox10:Cre; acta2:loxP-BFP-Stop-loxP-dsRed*

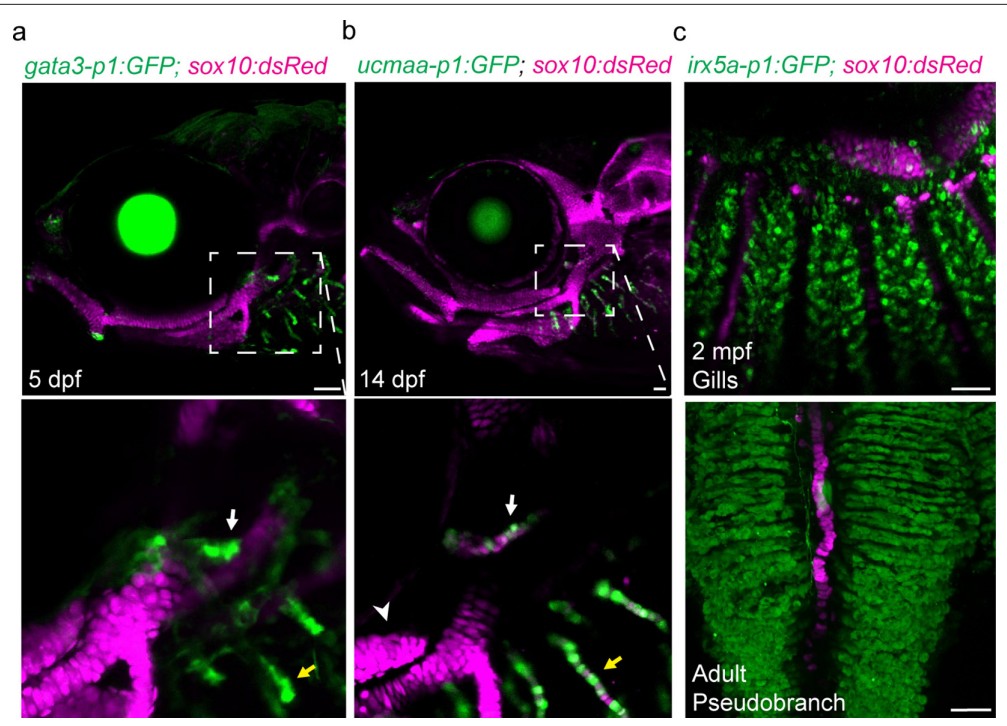

**Figure 2.** Shared regulatory program for pseudobranch and gill development. (**a-c**) In the pseudobranch (white arrows) and gill filaments (yellow arrows), *gata3-p1:GFP* labels growing buds, *ucmaa-p1:GFP* labels cellular cartilage (distinct from hyaline cartilage, arrowhead), and *irx5a-p1:GFP* labels pillar cells. *sox10:dsRed* labels cartilage for reference. Images in (**b**) and (**c**) are confocal projections, with magnified regions shown below in single sections for *gata3-p1:GFP* and *ucmaa-p1:GFP*. Scale bars, 50 μM.

or *gata3-p1:GFP* contributing to both the pseudobranch and gills (*Figure 3b and c*). Similar genetic dependency of the pseudobranch and gills further supports serial homology.

## Discussion

Our findings that both a cartilaginous and teleost fish have a mandibular gill-like pseudobranch suggest that the last common ancestor of jawed vertebrates did so as well, thus providing plausibility to the model that jaws evolved from a gill-bearing mandibular arch. The absence of a pseudobranch in extant agnathans (i.e. lamprey and hagfish) (*Cole, 1905*; *Mallatt, 1996*) suggests that either the pseudobranch arose along the gnathostome stem (i.e. prior to the divergence of cartilaginous and bony fishes), or that it was an ancestral feature of vertebrates that has been lost in cyclostomes. The latter would be analogous to loss of the hyoid hemibranch gill during teleost fish evolution (*Goodrich, 1930*; *Jollie, 1962*), consistent with our failure to observe gill filament gene expression or transgene activity in the hyoid arch of zebrafish.

Whereas our data clearly point to the filament systems of the pseudobranch and gills being serially homologous, the major skeletal bars supporting the jaws and gills (not to be confused with the gill filament cartilage) appear to develop largely independently from the filaments. Unlike the gill filaments, the zebrafish pseudobranch is not attached to a major skeletal bar. Conversely, the skeletal bars derived from the seventh arch of zebrafish lack gill filaments and instead anchor pharyngeal teeth, and no gill filaments were observed with the fossilized rostral-most gill bar of *M. walcotti* (*Morris and Caron, 2014*). In addition, *gata3* loss affects the pseudobranch and gill filaments but not the gill bars (*Sheehan-Rooney et al., 2013*). It is therefore possible that, rather than the pseudobranch evolving from an ancestral mandibular gill whose gill bar was transformed into the jaw skeleton, the pseudobranch arose independently after appearance of the jaw by co-option of a gill filament developmental program. While we demonstrate gill-like developmental potential of the mandibular arch in extant

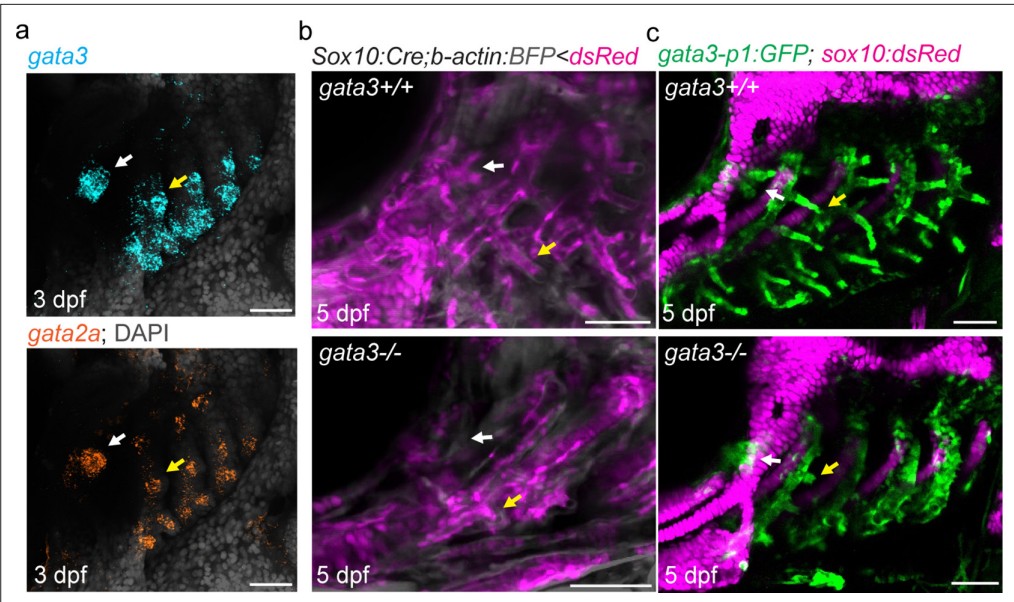

**Figure 3.** Pseudobranch and gill development requires *gata3* function. (**a**) Similar expression of *gata3* and *gata2a* in developing pseudobranch (white arrows) and gill regions (yellow arrows). (**b**) *Sox10:Cre; acta2:loxP-BFP-Stop-loxP-dsRed* labels Cre-converted dsRed+ neural crest-derived mesenchyme (magenta) and unconverted BFP+ epithelia (gray). (**c**) *gata3-p1:GFP* labels pseudobranch and gill filament buds, and *sox10:dsRed* labels cartilage. For both (**b**) and (**c**), 3/3 *gata3* mutants displayed reduced formation of the pseudobranch (white arrows) and gill filaments (yellow arrows), compared to 3 controls each. Scale bars, 50 µM.

The online version of this article includes the following figure supplement(s) for figure 3:

**Figure supplement 1.** Pseudobranch shares gene expression with gill filaments.

**Figure supplement 2.** The pseudobranch shares *irx5a-p1* pillar cell enhancer activity with gill filaments.

vertebrates, whether an ancestral mandibular gill bar gave rise to vertebrate jaws awaits more definitive fossil evidence.

## Materials and methods

### Key resources table

| Reagent type (species) or resource | Designation | Source or reference | Identifiers | Additional information |
|---|---|---|---|---|
| Gene (*Danio rerio*) | *ucmaa* | | Ensembl: ENSDARG00000027799 | |
| Gene (*Danio rerio*) | *gata3* | | Ensembl: ENSDARG00000016526 | |
| Gene (*Danio rerio*) | *gata2a* | | Ensembl: ENSDARG00000059327 | |
| Gene (*Danio rerio*) | *irx5a* | | Ensembl: ENSDARG00000034043 | |
| Genetic reagent (*Danio rerio*) | Tübingen | ZIRC | RRID:ZIRC_ZL57 | Wildtype strain of zebrafish |
| Genetic reagent (*Danio rerio*) | *Tg(fli1a:eGFP)^y1* | Lawson and Weinstein, 2002 | | |
| Genetic reagent (*Danio rerio*) | *Tg(sox10:kikGR)^el2* | Balczerski et al., 2012 | | |

*Continued on next page*

*Continued*

| Reagent type (species) or resource | Designation | Source or reference | Identifiers | Additional information |
|---|---|---|---|---|
| Genetic reagent (*Danio rerio*) | Tg(ucmaa_p1:GFP, cryaa:Cerulean)[el851] | **Fabian et al., 2022** | | |
| Genetic reagent (*Danio rerio*) | Tg(gata3_p1:GFP, cryaa:Cerulean)[el858] | **Fabian et al., 2022** | | |
| Genetic reagent (*Danio rerio*) | Tg(fgf10b:nEOS)[el865] | **Fabian et al., 2022** | | |
| Genetic reagent (*Danio rerio*) | Tg(–3.5ubb:loxP-STOP-loxP-mCherry)[el818] | **Fabian et al., 2020** | | |
| Genetic reagent (*Danio rerio*) | Tg(Mmu.Sox10-Mmu.Fos:Cre)[zf384] | **Kague et al., 2012** | | |
| Genetic reagent (*Danio rerio*) | Tg(actab2:loxP-BFP-STOP-loxP-dsRed)[sd27] | **Kobayashi et al., 2014** | | |
| Genetic reagent (*Danio rerio*) | Tg(–6.5kdrl:mCherry)[ci5] | **Proulx et al., 2010** | | |
| Genetic reagent (*Danio rerio*) | Tg(–5.0sox17:Cre-ERT2,myl7:DsRed)[sid1Tg] | **Hockman et al., 2017** | | |
| Genetic reagent (*Danio rerio*) | Tg(cdh1:mlanYFP)[xt17Tg] | **Cronan and Tobin, 2019** | | |
| Genetic reagent (*Danio rerio*) | gata3[b1075] | **Sheehan-Rooney et al., 2013** | | |
| Genetic reagent (*Danio rerio*) | Tg(irx5a-p1:GFP, cryaa:Cerulean)[el859] | This paper | | See Materials and Methods, Section Zebrafish Lines |
| Recombinant DNA reagent | PCS2FA-transposase | Tol2Kit | PUBMED: 17937395 396.pCS2-transposase | |
| Recombinant DNA reagent | pDestTol2AB2-irx5a-p1-E1B:GFP_pA | This paper | | See Materials and Methods, Section Zebrafish Lines |
| Sequence-based reagent | ucmaa RNAScope probe (*Danio rerio*); Channel 1 | ACD Bio | | |
| Sequence-based reagent | gata2a RNAScope probe (*Danio rerio*); Channel 1 | ACD Bio | | |
| Sequence-based reagent | gata3 RNAScope probe (*Danio rerio*); Channel 2 | ACD Bio | | |
| Commercial assay or kit | In-Fusion HD Cloning Plus | Takara | Takara:638,910 | |
| Commercial assay or kit | RNAScope Multiplex Fluorescent v2 Assay | ACD Bio | ACD Bio:323,100 | |
| Other | Draq5 nuclear dye | Abcam | Abcam:Ab108410 | See Materials and Methods, Section Imaging |

## Zebrafish lines

The Institutional Animal Care and Use Committee of the University of Southern California approved all animal experiments (Protocol 20771). Zebrafish lines include *Tg(fli1a:eGFP)[y1]* (**Lawson and Weinstein, 2002**); *Tg(–4.9sox10:kikGR)[el2]* (**Balczerski et al., 2012**); *Tg(ucmaa_p1:GFP, cryaa:Cerulean)[el851]*, *Tg(gata3_p1:GFP, cryaa:Cerulean)[el858]*, and *Tg(fgf10b:nEOS)[el865]* (**Fabian et al., 2022**); *Tg(–5.0sox17:Cre-ERT2,myl7:DsRed)[sid1Tg]* (**Hockman et al., 2017**); *Tg(cdh1:mlanYFP)[xt17Tg]* (**Cronan and Tobin, 2019**); *Tg(–3.5ubb:loxP-STOP-loxP-mCherry)[el818]* (**Fabian et al., 2020**); *Tg(Mmu.Sox10-Mmu.Fos:Cre)[zf384]* (**Kague et al., 2012**); *Tg(actab2:loxP-BFP-STOP-loxP-dsRed)[sd27]* (**Kobayashi et al., 2014**); *Tg(–6.5kdrl:mCherry)[ci5]* (**Proulx et al., 2010**); and *gata3[b1075]* (**Sheehan-Rooney et al., 2013**). To generate *Tg(irx5a-p1:GFP, cryaa:Cerulean)[el859]*, we synthesized the intergenic peak associated with *irx5a* (chr7:35838071–35838577) using iDT gBlocks, cloned it into a modified pDestTol2AB2 construct containing the E1b minimal promoter, GFP, polyA, and the lens-specific *cryaa:Cerulean* marker using in-Fusion cloning (Takara Bio). We injected plasmid and Tol2 transposase RNA (5–10 ng/µL each) into

one-cell stage zebrafish embryos and screened for founders at adulthood based on lens CFP expression in progeny. Two independent germline founders were identified that showed activity in gill pillar cells.

## Histology

Adult fish were fixed in 4% paraformaldehyde for 1 hr at 25°C followed by dissection of the gills and further fixation in 4% paraformaldehyde for 1 hr at 25°C. For pseudobranch dissection, adults were fixed in 4% paraformaldehyde at 4°C for 3 days prior to dissection. Alcian Blue staining was performed on whole tissue as previously described (*Paul et al., 2016*). Samples were imaged using a Leica DM2500 microscope. Image levels were adjusted in Adobe Illustrator.

## Photoconversion-based lineage tracing

To photoconvert mesenchyme in *sox10:kikGR; fli1a:GFP* fish at 1.5 dpf, we used the ROI function in ZEN software on a Zeiss LSM800 confocal microscope to expose dorsal first or second arch mesenchyme to UV light for 20 s. Imaging confirmed successful and specific photoconversion of kikGR from green to red fluorescence in the intended region. At 3.5 dpf, confocal imaging was used to assess contribution of photoconverted cells to pseudobranch mesenchyme. We included *fli1a:GFP* to help in identification of the pseudobranch bud. For *fgf10b:nEOS* photoconversion, we used the ROI function to expose nEOS-high expressing cells in the first or third pharyngeal pouches to UV light for 60 s, with immediate confocal imaging confirming intended photoconversion of nEOS from green to red fluorescence. At 5 dpf, confocal imaging was used to assess contribution of photoconverted cells to pseudobranch and gill epithelia. To confirm that nEOS-high expressing cells in the *fgf10b:nEOS* line were of endodermal original, we crossed these onto the *sox17:CreERT2; ubb:loxP-Stop-loxP-mCherry* transgenic background and treated embryos with 4-hydroxytamoxifen (Sigma) at 6.5 hpf to induce Cre recombination. We then imaged on the confocal microscope at 1.5 dpf to visualize co-localization of nEOS and mCherry. All results were independently confirmed in at least three animals.

## In situ hybridization

We performed in situ hybridization on whole embryos at 3 and 5 dpf and on paraffin sections from adult zebrafish heads using RNAscope probes synthesized by Advanced Cell Diagnostics in channel 1 (*ucmaa*, *gata2a*) and channel 2 (*gata3*). Samples were prepared by fixation in 4% paraformaldehyde overnight. Embryos were dehydrated in methanol and stored overnight before proceeding with the RNAScope Assay for Whole Zebrafish Embryos as described in the manufacturer's protocols. Following fixation, the pseudobranch was dissected and mounted in 0.2% agarose in molds. Once solidified, agarose chips containing the pseudobranch were cut out of the mold, dehydrated, embedded in paraffin, and 5 µm sections were collected using a Shandon Finesse Me+ microtome (cat. no. 77500102). Paraformaldehyde-fixed paraffin-embedded sections were deparaffinized, and the RNAscope Fluorescent Multiplex V2 Assay was performed according to manufacturer's protocols using an ACD HybEZ Hybridization oven. In situ patterns were confirmed in at least three independent animals, with exception of the *ucmaa* in situ that was performed on three separate sections of the same animal.

## Imaging

Images of whole-mount or section fluorescent in situ hybridizations and live transgenic fish were captured on a Zeiss LSM800 confocal microscope using ZEN software. For adult imaging of the *irx5a-p1:GFP* reporter, whole animals were euthanized and the pseudobranch and gills dissected out. The tissue was stained with Draq5 nuclear dye (Abcam) for 20 min to help identify pillar cells. Reported expression patterns for enhancer lines were confirmed in at least five animals.

## Acknowledgements

We thank Megan Matsutani for fish care, Johann Eberhart and Mary Swartz for the *gata3* mutant fish, and Keith Cheng, Jean Copper, and Daniel Vanselow for providing the original images related to Figure 1b. We used https://biorender.com/ to create Figure 1a.

## Additional information

### Funding

| Funder | Grant reference number | Author |
|---|---|---|
| National Institute of Dental and Craniofacial Research | 5R35DE027550 | J Gage Crump |
| National Institute of Dental and Craniofacial Research | 5K99DE029858 | Peter Fabian |
| National Institute of Dental and Craniofacial Research | 1F31DE030706 | Mathi Thiruppathy |
| Royal Society | UF130182 | J Andrew Gillis |
| Royal Society | URF\R\191007 | J Andrew Gillis |

The funders had no role in study design, data collection and interpretation, or the decision to submit the work for publication.

### Author contributions

Mathi Thiruppathy, Conceptualization, Formal analysis, Investigation, Methodology, Writing - original draft, Writing – review and editing; Peter Fabian, Investigation; J Andrew Gillis, Conceptualization, Writing – review and editing; J Gage Crump, Formal analysis, Funding acquisition, Supervision, Writing - original draft, Writing – review and editing

### Author ORCIDs

Mathi Thiruppathy (iD) http://orcid.org/0000-0002-6337-3256
J Andrew Gillis (iD) http://orcid.org/0000-0003-2062-3777
J Gage Crump (iD) http://orcid.org/0000-0002-3209-0026

### Ethics

This study was performed in strict accordance with the recommendations in the Guide for the Care and Use of Laboratory Animals of the National Institutes of Health. The Institutional Animal Care and Use Committee of the University of Southern California approved all animal experiments (Protocol 20771).

### Decision letter and Author response

Decision letter https://doi.org/10.7554/eLife.78170.sa1
Author response https://doi.org/10.7554/eLife.78170.sa2

## Additional files

### Supplementary files

• MDAR checklist

### Data availability

The current manuscript contains solely images, so no data have been generated for this manuscript. The n number for each image type is clearly stated in the manuscript.

The following previously published dataset was used:

| Author(s) | Year | Dataset title | Dataset URL | Database and Identifier |
|---|---|---|---|---|
| Crump G, Fabian P, Tseng K, Chen H | 2021 | Single-cell profiling of cranial neural crest diversification across a vertebrate lifetime | https://www.ncbi.nlm.nih.gov/geo/query.cgi?acc=GSE178969 | NCBI Gene Expression Omnibus, GSE178969 |

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
