## [Editor Report]

This is an interesting and important paper that investigates pseudobranch development in zebrafish in the context of seeking evidence for a proposed gill arch origin for the vertebrate jaw. It provides data that supports that the pseudobranch is derived from the mandibular arch and that the pseudobranch is a segmental homolog of the gills providing strong support for the classic gills-to-jaws hypothesis.

---

## [Decision Letter]

**Decision letter after peer review:**

Thank you for submitting your article "Gill developmental program in the teleost mandibular arch" for consideration by *eLife*. Your article has been reviewed by 2 peer reviewers, and the evaluation has been overseen by a Reviewing Editor and Marianne Bronner as the Senior Editor. The following individual involved in the review of your submission has agreed to reveal their identity: Craig T Miller (Reviewer #2).

In this manuscript, the authors investigate pseudobranch development in zebrafish in the context of seeking evidence for a proposed gill arch origin for the vertebrate jaw. They provide data that supports that the pseudobranch is derived from the mandibular arch and that the pseudobranch is a segmental homolog of the gills. The reviewers agree that this work is potentially appropriate for publication in *eLife* if certain concerns can be addressed.

1. The first arch kikGR fate mapping example in Figure 1e is of poor resolution and it's difficult to see the cellular composition of the forming pseudobranch. (by contrast, the Sox10Cre:b-actin:BFP)

2. Similarly for Figure 3S2b (the 13 dpf irx5a-p1 figure) is that the best representative figure? This is a weak data point with only a single blip of GFP positive signal which may or may not even be a cell.

3. The fgf10b:nEOS does not appear to be truly endodermal specific. Thus, it's not clear whether the magenta label in Figure 1f came from the first pouch as claimed, or alternatively from non-endodermal fgf10b expressing cells. Do the authors have data showing sox17:CreER; ubb:stop-mCherry tamoxifen labeled magenta cells in the pseudobranch? Alternatively, can the authors show an example of photoconverting nEOS in the fgf10b:nEOS line that is more localized onto the first pharyngeal pouch (the example in Figure 1F appears to include photoconverted cells both posterior but especially more ventral than the first pharyngeal pouch).

4. It seems that in this group's recent 2022 Nat. Comm. scRNA-seq paper, pseudobranch cells might also have been sequenced. Can the authors ask whether pseudobranch cells in their existing scRNA-seq data cluster with different gill cell clusters? E.g. perhaps gill cells are hox+ (and gata3+, ucmaa+, or irx5a+) while pseudobranch cells are hox- (and gata3+, ucmaa+, or irx5a+)? Perhaps the sequencing depth is not deep enough to answer this question, or perhaps the authors have no markers that distinguish pseudobranch cells from gill cells and this approach would not work. But it seems possible that the scRNA-seq data could provide additional evidence for serial homology of the pseudobranch and gills.

5. Are there gill-like structures derived from the second arch in teleosts? The authors mention the hemibranch derived from the hyoid arch in cartilaginous fishes. Is there a similar structure in any teleost? If there are no arch 2 gill-like structures, how this fits into the gill origin of jaw hypothesis should be discussed. Also, there is no discussion of jawless fish and how the considerable literature from species such as lamprey relates to the findings presented here. Do agnathans develop a pseudobranch? Addressing both of these points would help a general audience.

6. Better care should be taken with the word homology, and the distinction between serial homology vs historical or functional homology should be made clear. It would be helpful to define for a general audience what is meant by serial homology.

7. The first sentence of the discussion is not fully supported by the data. More species need to be included to make this claim. This statement should be tempered.

Other points:

In the figures, the font size of text and figure panels should be increased throughout for clarity.

In Figure 1A, the jaw and hyoid skeleton is cartooned in gray, not just the hyoid, so in the Figure 1A legend, "jaw support skeleton" should be more accurately changed to "jaw and jaw support skeleton". The uninterrupted medial gray bar should be removed from this diagram as it's neither anatomically accurate nor relevant for the images/data in this paper. Likewise, the gray blob in front of the pterygoid process should either be removed or extended posteriorly to more accurately represent the neurocranium.

In the Figure 1D legend, one of the two "in green" should be removed from this sentence: "In green, fli1a:GFP labels the vasculature and neural crest-derived mesenchyme for reference in green, and unconverted sox10:kikGR cells also labels mesenchyme."

The Figure 1F legend is mislabeled as "e".

The Figure 3A legend should define the white arrow and the yellow arrow separately.

The Figure 1S1D legend indicates that pouches are numbered, but they are not numbered in the figure. Either numbers should be added or the legend corrected.

In Figure 3S1, labeling the gill buds with yellow arrows as in the other figures would be helpful and make the presentation more consistent.

p. 4, Reference(s) should be provided to support this claim: "The pseudobranch is an epithelial structure located just behind the eye of most fishes that regulates ocular blood pressure."

p. 7 "Gata3 and Gata2a" to "gata3 and gata2a (italicized)"?

*Reviewer #2 (Recommendations for the authors):*

Suggestions to strengthen the science are listed below:

1. The first arch kikGR fate mapping example in Figure 1e is of lowish resolution and difficult to see the cellular composition of the forming pseudobranch. It is listed that an n of 6 was done here. In contrast, the example shown in Figure 1S1A with the Sox10Cre:b-actin:BFP

2. The fgf10b:nEOS does not appear to be truly endodermal specific. Thus, it's not clear whether the magenta label in Figure 1f came from the first pouch as claimed, or alternatively from non-endodermal fgf10b expressing cells. Do the authors have data showing sox17:CreER; ubb:stop-mCherry tamoxifen labeled magenta cells in the pseudobranch? Alternatively, can the authors show an example of photoconverting nEOS in the fgf10b:nEOS line that is more localized onto the first pharyngeal pouch (the example in Figure 1F appears to include photoconverted cells both posterior but especially more ventral than the first pharyngeal pouch).

3. It seems that in this group's recent 2022 Nat. Comm. scRNA-seq paper, pseudobranch cells might also have been sequenced. Can the authors ask whether pseudobranch cells in their existing scRNA-seq data cluster with different gill cell clusters? E.g. perhaps gill cells are hox+ (and gata3+, ucmaa+, or irx5a+) while pseudobranch cells are hox- (and gata3+, ucmaa+, or irx5a+)? Perhaps the sequencing depth is not deep enough to answer this question, or perhaps the authors have no markers that distinguish pseudobranch cells from gill cells and this approach would not work. But it seems possible that the scRNA-seq data could provide additional evidence for serial homology of the pseudobranch and gills.

Suggestions to improve the manuscript:

4. In the figures, the font size of text and figure panels should be increased throughout for clarity.

5. In Figure 1A, the jaw and hyoid skeleton is cartooned in gray, not just the hyoid, so in the Figure 1A legend, "jaw support skeleton" should be more accurately changed to "jaw and jaw support skeleton". The uninterrupted medial gray bar should be removed from this diagram as it's neither anatomically accurate nor relevant for the images/data in this paper. Likewise, the gray blob in front of the pterygoid process should either be removed or extended posteriorly to more accurately represent the neurocranium.

6. In the Figure 1D legend, one of the two "in green" should be removed from this sentence: "In green, fli1a:GFP labels the vasculature and neural crest-derived mesenchyme for reference in green, and unconverted sox10:kikGR cells also labels mesenchyme."

7. The Figure 1F legend is mislabeled as "e".

8. The Figure 3A legend should define the white arrow and the yellow arrow separately.

9. The Figure 1S1D legend indicates that pouches are numbered, but they are not numbered in the figure. Either numbers should be added or the legend corrected.

10. In Figure 3S1, labeling the gill buds with yellow arrows as in the other figures would be helpful and make the presentation more consistent.

11. p. 4, Reference(s) should be provided to support this claim: "The pseudobranch is an epithelial structure located just behind the eye of most fishes that regulates ocular blood pressure."

12. p. 6, "Zebrafish mutant for gata3 fails to form gill buds": fails to fail, as zebrafish is plural here?

13. p. 7 "Gata3 and Gata2a" to "gata3 and gata2a (italicized)"?

14. p. 7 "…jaws evolved from a mandibular gill." Change to "…from a mandibular gill support", "…from a mandibular segment containing a gill", "…from a gill-bearing mandibular arch"? I don't think of a gill as containing the skeletal support element, so the wording as-is will be confusing to some.

No issues with the availability of data or reagents were identified.

---

## [Author Response]

1. The first arch kikGR fate mapping example in Figure 1e is of poor resolution and it's difficult to see the cellular composition of the forming pseudobranch. (by contrast, the Sox10Cre:b-actin:BFP)

We have repeated the *sox10:kikGR* imaging at better resolution and now show clearer data in Figure 1e. This has also led to an increase in n number for both first and second arch conversion.

2. Similarly for Figure 3S2b (the 13 dpf irx5a-p1 figure) is that the best representative figure? This is a weak data point with only a single blip of GFP positive signal which may or may not even be a cell.

The purpose of including an image of the irx5a-p1:GFP transgenic line at 13 dpf is to highlight the very earliest timepoint at which we see signal in the pseudobranch. While it is true that there are very few GFP+ cells at this stage, this is consistent across multiple animals. We now include a second example from this stage with slightly stronger contribution in Figure 3—figure supplement 2b. As can be seen in Figure 2c and Figure 3—figure supplement 2c, the number of irx5a-p1:GFP+ cells in the pseudobranch greatly increases by late juvenile and adult stages.

3. The fgf10b:nEOS does not appear to be truly endodermal specific. Thus, it's not clear whether the magenta label in Figure 1f came from the first pouch as claimed, or alternatively from non-endodermal fgf10b expressing cells. Do the authors have data showing sox17:CreER; ubb:stop-mCherry tamoxifen labeled magenta cells in the pseudobranch? Alternatively, can the authors show an example of photoconverting nEOS in the fgf10b:nEOS line that is more localized onto the first pharyngeal pouch (the example in Figure 1F appears to include photoconverted cells both posterior but especially more ventral than the first pharyngeal pouch).

We performed a lineage trace of *sox17:CreERT2*-labelled endoderm and show contribution to both gill and pseudobranch *cdh1:mlanYFP*+ epithelia (Figure 1—figure supplement 1d). We also performed more specific photoconversion of *fgf10b:nEOS* in just the first pouch and show similar contribution to pseudobranch epithelia (new Figure 1f, old exampled moved to Figure 1figure supplement 1d). These new experiments now better support contribution of first pouch endoderm to the pseudobranch epithelium.

4. It seems that in this group's recent 2022 Nat. Comm. scRNA-seq paper, pseudobranch cells might also have been sequenced. Can the authors ask whether pseudobranch cells in their existing scRNA-seq data cluster with different gill cell clusters? E.g. perhaps gill cells are hox+ (and gata3+, ucmaa+, or irx5a+) while pseudobranch cells are hox- (and gata3+, ucmaa+, or irx5a+)? Perhaps the sequencing depth is not deep enough to answer this question, or perhaps the authors have no markers that distinguish pseudobranch cells from gill cells and this approach would not work. But it seems possible that the scRNA-seq data could provide additional evidence for serial homology of the pseudobranch and gills.

We thank the reviewer for this excellent suggestion. Unfortunately, we have analyzed our scRNAseq data at 60 dpf but find no clear segregation of hox+ and hox- cells within gill cell types (*ncam3*+ pillar cells, *ucmaa*+ gill filament chondrocytes, *fgf10a*+ gill progenitors). The only Hox genes we detect at this stage are *hoxa3a* and *hoxb3a*, yet these do not co-localize in any clear subset of gill cells. It may be that Hox expression has largely been extinguished during these later differentiation stages, and/or that gill cells greatly outnumber pseudobranch cells, thus making it difficult to detect a distinct hox- pseudobranch cluster.

**Author response image 1. sa2fig1:** 

5. Are there gill-like structures derived from the second arch in teleosts? The authors mention the hemibranch derived from the hyoid arch in cartilaginous fishes. Is there a similar structure in any teleost? If there are no arch 2 gill-like structures, how this fits into the gill origin of jaw hypothesis should be discussed. Also, there is no discussion of jawless fish and how the considerable literature from species such as lamprey relates to the findings presented here. Do agnathans develop a pseudobranch? Addressing both of these points would help a general audience.

We now cite references that a hyoid hemibranch is present in non-teleost but not teleost fishes, suggesting it was lost along the teleost lineage. This is consistent with our observed lack of gill filament gene expression and transgene activity in the zebrafish hyoid arch. We also now cite references for a hyoid hemibranch and more posterior gills in lamprey, yet a lack of pseudobranch in either lamprey or hagfish.

pp. 2-3: “The second (hyoid) arch also forms a hemibranch (one row of gill filaments) in the jawless lamprey fish (Dohrn, 1882; Gaskell, 1908), in cartilaginous and various non-teleost fishes (e.g. coelacanth, lungfishes, sturgeon, gar), but not in teleost fishes (Goodrich, 1930; Jollie, 1962).”

pp. 3: “However, extant agnathans (the cyclostomes lamprey and hagfish) lack a mandibular gill (Cole, 1905; Mallatt, 1996)…”

pp. 6-7: “The absence of a pseudobranch in extant agnathans (i.e. lamprey and hagfish) (Cole, 1905; Mallatt, 1996) suggests that either the pseudobranch arose later along the gnathostome lineage, or that it was lost along the cyclostome lineage. The latter would be analogous to loss of the hyoid hemibranch gill during teleost fish evolution (Goodrich, 1930; Jollie, 1962), consistent with our failure to observe gill filament gene expression or transgene activity in the hyoid arch of zebrafish.”

6. Better care should be taken with the word homology, and the distinction between serial homology vs historical or functional homology should be made clear. It would be helpful to define for a general audience what is meant by serial homology.

We now define serial homology in the Introduction and are consistent with use of “serial homology” throughout, including in the Discussion as suggested.

p.3: “Another important question is whether the pseudobranch and gills can be considered serially homologous, i.e. representing morphologically related structures that arise through shared developmental and genetic mechanisms.”

7. The first sentence of the discussion is not fully supported by the data. More species need to be included to make this claim. This statement should be tempered.

We have modified this sentence to stress that our conclusions are based on close examination of only two species.

P 6: “Our findings that both a cartilaginous and teleost fish have a mandibular gill-like pseudobranch suggest that the last common ancestor of jawed vertebrates did so as well, thus providing plausibility to the model that jaws evolved from a mandibular gill.”

Other points:In the figures, the font size of text and figure panels should be increased throughout for clarity.

We have increased font size throughout the figures.

In Figure 1A, the jaw and hyoid skeleton is cartooned in gray, not just the hyoid, so in the Figure 1A legend, "jaw support skeleton" should be more accurately changed to "jaw and jaw support skeleton". The uninterrupted medial gray bar should be removed from this diagram as it's neither anatomically accurate nor relevant for the images/data in this paper. Likewise, the gray blob in front of the pterygoid process should either be removed or extended posteriorly to more accurately represent the neurocranium.

The Figure 1a schematic and legend have been changed as suggested.

In the Figure 1D legend, one of the two "in green" should be removed from this sentence: "In green, fli1a:GFP labels the vasculature and neural crest-derived mesenchyme for reference in green, and unconverted sox10:kikGR cells also labels mesenchyme."

Corrected.

The Figure 1F legend is mislabeled as "e".

Corrected.

The Figure 3A legend should define the white arrow and the yellow arrow separately.

Corrected.

The Figure 1S1D legend indicates that pouches are numbered, but they are not numbered in the figure. Either numbers should be added or the legend corrected.

We have added pouch numbers to the figures.

In Figure 3S1, labeling the gill buds with yellow arrows as in the other figures would be helpful and make the presentation more consistent.

Figures S2 and S3 have been modified to include yellow arrows indicating gill buds as suggested. We have also double-checked that each figure contains white arrows for pseudobranches and yellow arrows for gills.

p. 4, Reference(s) should be provided to support this claim: "The pseudobranch is an epithelial structure located just behind the eye of most fishes that regulates ocular blood pressure."

By examining Jollie, 1962 and other anatomical literature, it is apparent that the function of the pseudobranch is not completely resolved. We therefore edit the text to reflect the two current theories behind its function.

“The pseudobranch is an epithelial structure located just behind the eye that has been proposed to regulate ocular blood pressure and/or have an endocrine function (Jollie, 1962).”

p. 7 "Gata3 and Gata2a" to "gata3 and gata2a (italicized)"?

Corrected.